# Professional Obstacles to Anaesthesiology Practice in Punjab, Pakistan: Qualitative Study of Consultant Anaesthesiologists’ Perspectives

**DOI:** 10.3390/ijerph192013427

**Published:** 2022-10-18

**Authors:** Sumbal Shahbaz, Rubeena Zakar, Florian Fischer, Natasha Howard

**Affiliations:** 1Department of Public Health, University of the Punjab, Lahore 54590, Pakistan; 2Saw Swee Hock School of Public Health, National University of Singapore, Singapore 119077, Singapore; 3Department of Allied Health Sciences, Superior University, Lahore 54000, Pakistan; 4Institute of Public Health, Charité—Universitätsmedizin Berlin, 10117 Berlin, Germany; 5Bavarian Research Center for Digital Health and Social Care, Kempten University of Applied Sciences, 87437 Kempten, Germany; 6Department of Global Health & Development, London School of Hygiene and Tropical Medicine, London 400706, UK

**Keywords:** anaesthesia, workforce challenges, qualitative research, Pakistan

## Abstract

Limitations in the global anaesthesia workforce contribute to the emigration of skilled anaesthesiologists from lower-income to higher-income countries, jeopardizing workforce balance and patient outcomes in Pakistan. This study aimed to explore the challenges experienced by anaesthesiologists in Punjab, Pakistan’s most populous province, and the potential changes to encourage their retention. We conducted a qualitative study, conducting semi-structured interviews with 25 purposively sampled consultant anaesthesiologists working in Punjab and analysing data thematically. Reported professional challenges and reasons consultant anaesthesiologists chose to work abroad differed between public and private sectors, each sector providing distinct challenges that compromised anaesthesia workforce numbers and quality. Key concerns were security, promotion/incentive structures, and gender inequalities in public hospitals versus inadequate salary and facilities, surgeon dependency, and the lack of out-of-theatre practice in private hospitals that minimized the scope and earnings of anaesthesiologists within Pakistan. Our findings help contextualise Pakistan’s anaesthesia workforce crisis, indicating public-sector improvements could include increasing security in hospital premises, performance-based incentives, and qualification-dependent promotion, while private-sector improvements could include decreasing surgeon dependency, fixing salary percentages by surgical case, and encouraging direct patient-anaesthesiologist relationships. National and subnational interventions to promote anaesthesiology, along with public awareness campaigns, could additionally raise its profile and encourage retention.

## 1. Introduction

Limitations in the global anaesthesia workforce have disproportionately affected low- and middle-income countries such as Pakistan, as skilled professionals emigrate to high-income countries for better opportunities [1]. This one-way migration jeopardizes clinical workforce balance and patient outcomes by shifting tasks to often less rigorously trained non-physician anaesthesiologists. 

In South Asia, anaesthesiology is not a preferred clinical specialty due to lack of recognition, surgeon dependence, professional stress, scarce research funding, and medico-legal issues [2]. The World Health Organization (WHO) reported that six of seven South Asian countries lack anaesthesiologists due to emigration, national maldistribution, insufficient specialist training, and increased demand [3]. For example, only 10% of sub-district hospitals in India have anaesthesiologists, while skilled anaesthesia staff in Pakistan and Bangladesh are maldistributed towards urban areas [4]. In these countries, anaesthesiologists are underappreciated by both the public and medical colleagues in other specialties despite their crucial intraoperative and postoperative roles [5]. Thus, 25-30% of anaesthesiologist trained in Pakistan work in other countries, while 37% of Sri Lankan anaesthesiologists are in the United Kingdom and United States [4].

The public has little knowledge of anaesthesia procedures and personnel in resource-constrained countries [6] and their roles intra and postoperatively are poorly understood [7]. In India and Pakistan, 58% and 49% of people, respectively, had no idea who would anesthetize them or the associated risks [8,9]. The limited appreciation for this speciality, dependence on surgeons, lower wages and ultimately lower job satisfaction causes many anaesthesiologists to emigrate and subsequent national ‘brain drain’ [10,11]. Although the COVID-19 pandemic has highlighted the importance and versatility of this profession, its future in lower-income countries remains controlled by its relationship with surgery [12].

Data on anaesthesiology in Pakistan are lacking and no research has highlighted anaesthesiologists perceived challenges. Many anaesthesiologists in Pakistan hold fulltime public sector employment while also working formally or informally in private health facilities, primarily due to financial reasons and the lack of qualified professionals in country [13,14]. Anaesthesiologists thus receive a monthly public-sector salary and taking additional surgical assignments does not affect their earnings, so ‘topping-up’ with private sector work provides additional income. Efforts to restrict private practice of public sector clinicians have failed due to poor private sector regulation [15] and dual public–private employment remains common for many medical specialities [16]. Research is needed on why fewer medical professionals opt for this speciality and why skilled anaesthesiologists continue to emigrate from Pakistan.

This study thus aimed to explore reported challenges among anaesthesiologists working in Punjab province, the most populous province in Pakistan, and potential policy or practice changes that could improve working conditions and opportunities for this speciality within the country.

## 2. Methods

### 2.1. Study Design

We conducted a qualitative study, drawing on semi-structured interviews with skilled anaesthesiologists in public and private practice in Punjab province. Our research question was: “What are the major reported challenges to practicing anaesthesiology in Punjab, Pakistan?” 

### 2.2. Participant Selection

We used purposive and snowball sampling to recruit anaesthesiologists with postgraduate qualifications (i.e., doctor of medicine (MD), Member/Fellow of the College of Physicians and Surgeons (MCPS/FCPS)) working in public hospitals in Punjab (i.e., Township/‘Tehsil’ Headquarters Hospital (THQ), District Headquarters Hospital (DHQ), teaching hospital) as consultants, registrars, senior registrars, or department heads, with at least one year of professional experience and aged under 60 years old. An initial list of potential participants in teaching hospitals were contacted through their official numbers, available online, and asked to refer one or more potential DHQ/THQ participants [17,18,19]. None refused to participate in the study.

### 2.3. Data Collection

We developed a semi-structured interview guide informed by the literature and expert consultation within University of the Punjab, which covered demographics (e.g., age, position, years of experience), challenges in practicing anaesthesia, challenges in acceptance from colleagues and patients, management and monetary issues, and interviewee suggestions for addressing challenges described. The guide allowed for both discussion of deductive concerns and emergence of unexpected issues.

SS contacted potential interviewees on their work phone to explain the study and invite their participation. SS conducted in-person semi-structured interviews in Urdu from June to November 2021, after audio-recording verbal informed consent, as government employees are generally unwilling to sign consent forms in case they might pose a job risk. Interviews took 40–50 min and were audio-recorded with participant consent. Nine did not agree to audio recording due to security and administrative sensitivities, so SS took detailed notes. SS translated and transcribed notes and audio files into English, which were then checked by RZ. SS and RZ determined data saturation was achieved when no new ideas or concepts were emerging from interviews [20,21]. We safeguarded participant anonymity and confidentiality by ensuring they chose interview times and locations, using identification codes instead of names on all outputs, deleting audio files after transcription, and storing transcripts in a password-protected hard drive only accessible by the research team.

### 2.4. Data Analysis

The Department of Public Health, Institutional Review Board at University of the Punjab in Pakistan provided ethical approval (1456/Acad.; 22 February 2020). Informed written consent was taken from each participant prior to each interview. SS analysed data thematically using Braun and Clarke’s six phases [22], i.e., reading and re-reading; initial noting; developing themes; searching for connections across themes; moving to the next subtheme; and looking for patterns across subthemes. NH reviewed themes and subthemes and co-authors agreed final interpretations [23]. Reporting adheres to COREQ criteria [24].

### 2.5. Ethics

The Department of Public Health Institutional Review Board at University of the Punjab in Pakistan provided ethical approval (1456/Acad.; 22 February 2020). 

## 3. Results

### 3.1. Participants’ Characteristics and Analytical Themes

Table 1 shows 25 participants, of whom 10 (40%) worked in public sector teaching hospitals (TH), 8 (32%) in District Headquarter (DHQ) hospitals, and 7 (28%) in Tehsil Headquarter (THQ) hospitals. Approximately two-thirds (68%) were men, averaging 44 years old and 11 years of experience, and 8 (32%) were women, averaging 39 years old and 7 years’ experience. Consultants averaged 14 years’ experience in teaching hospitals (i.e., 15 for men, 12 for women), 7 years in DHQ (10 for men, 3.5 for women), and 6 years in THQ (6 for men, 4 for women). All participants worked full or part-time in public hospitals and ‘on-call’ for private hospitals in evenings, so they could provide first-hand experience of both settings.

We separated findings by public (i.e., government operated) or private sector, as issues in both were sufficiently distinct and participants had experience of both. Major obstacles identified in the public sector were personal security, facilities for women anaesthesiologists, differentiation between consultant and specialist by the hospital administration, payment and incentives, and privatising public hospitals in evenings. Major private-sector obstacles were inadequate salaries and facilities, surgeon dependency, lack of out-of-theatre practice, fee fixation, and surgical hierarchy.

### 3.2. Practice Hurdles in Public Sector Health Facilities

#### 3.2.1. Security

Participants described security as the primary concern in all public hospitals in Pakistan. Security was particularly concerning in peripheral (i.e., smaller non-central urban, semi-urban, and rural) areas where equipment and medication were available to administer anaesthesia, but anaesthesiologists could not risk of anesthetizing patients for serious surgeries due to fears for their own security, e.g., being beaten or injured by patients’ accompanying family for any adverse outcomes. 

“*Why would I put my life in danger by anesthetizing a patient who is already in shock with insufficient blood in hand and a mob of patient’s family outside theatre ready to rip me off in case of his demise*”.(QH5)

Half of participants reported that patients in peripheral areas were often from low socioeconomic backgrounds with unidentified comorbidities (e.g., high blood pressure, diabetes, cardiac abnormalities) and arrived on the day of surgery due to insufficient inpatient beds. Superficial anaesthetic pre-operative checks are done ten minutes before surgery, making it impossible to handle serious cases due to insufficient background knowledge. All commented that focusing on patients’ wellbeing was difficult when their own was in danger, thus they preferred providing anaesthesia for relatively less risky day surgeries or caesarean sections. A DHQ participant noted:

“*Government has appointed a few police sergeants outside each hospital which cannot control violent and angry attendants—especially when there are hundreds of them out there. So, in peripheral setups, best is to take non-serious or ASA 1 [American Society of Anaesthesiologist Classification] cases. I know it’s not good for patients, but it’s not good to risk healthcare team lives either*”.(DH2)

Teaching hospitals were somewhat better, as patients for elective surgeries are optimised before surgery, while for emergency patients, police were on standby to control attending relatives, and facilities had emergency exit doors for staff in case of trouble. A few claimed that security concerns in secondary hospitals were a poor excuse for referring patients to teaching hospitals as management were not interested in the wellbeing of healthcare staff or the public in any hospital. They suggested that not allowing patients’ relatives to enter surgical floors would resolve security issues.

“*There is no check and balance anywhere, just we don’t have lame excuse to refer it to some other hospital. So, we save patients life while peripheral workforce prefers to save their own*”.(TH2)

#### 3.2.2. Gender Inequalities

Women are becoming indispensable to the anaesthesia workforce. However, most do not enter the workforce after postgraduate studies. While some reasons were complex, one was unacceptably low wages. As women are not considered family breadwinners in Pakistan, they only worked if salary or facilities were sufficiently good. A senior consultant shared her experience: 

“*When I was an MCPS student 20 years back, I used to get 3 k for a case, and immediately used to go for it, but now after being an FCPS having 20 years of experience why would I go for 5 or 10 K for case where surgeon having equal qualification is getting in 100–500 K. Even, I have to start before him and finish after him due to pre- and post-ops—so why don’t I spend quality time with my family instead?*”.(TH8)

For some women, it was not salary but facilities that deterred them. For example, the lack of good-quality and safe 24 h childcare or safe late-night transportation.

“*Transport facilities must be provided to female staff in every hospital especially in night shift when moving alone in local transport or private taxis is generally not considered safe in our country*”.(TH7)

Others described poor security within hospitals, complaining that every theatre had an office for female surgeons to rest and refresh themselves after surgery, but no such spaces were provided for female anaesthesiologists who had to sit with male colleagues whether or not they were comfortable with this. They advocated for rooms for female anaesthesiologists, particularly when working night shifts, inside or near theatres with a security guard present. As almost 70% of anaesthesiology consultants in Punjab are women, due to male anaesthesiologists moving abroad, women could be better integrated within the system by improving their perceived safety.

#### 3.2.3. Experiential Seniority Outranking Qualifications

Several participants noted that in Pakistan, all anaesthesia personnel with postgraduate degrees (e.g., DA (Diploma of Anaesthesia) or FCPS) are considered consultants and prioritised according to their hospital experience regardless of their degree, which was confusing and needs to be corrected. Teaching hospital participants insisted on not calling diploma-holders’ consultants, noting they should be referred to as specialists to overcome this confusion. Almost all participants agreed that those who are worthy should be given due rights and positions or the system would continue losing qualified personal because of perceived disrespect and poor treatment/favouritism by unqualified seniors.

“*Government made rules to move every MCPS and DA consultant to peripheral hospitals and FCPS/MS consultants to stay in tertiary care and train other residents. But unfortunately, in many tertiary care hospitals DA consultants are leading […] and FCPS consultants have to work under them, which creates chaos, ultimately leading FCPS consultants to give up and move out of the system. So, the differentiation in status according to specification of degree is inevitable now*”.(TH1)

“*In several teaching hospitals, despite having numerous consultant anaesthesiologists in country, non-consultants are given authority and only support underqualified people like themselves. [This] creates challenges for more qualified professionals who ultimately leave the system due to the harsh and exhausting work environment*”.(TH4)

#### 3.2.4. Pay and Incentives

All participants expressed concerns about pay grades and lack of incentives, insisting that doctors globally are highly paid professionals, with surgeons and anaesthesiologists entitled to the highest salaries. However, in Pakistan, they received equivalent salaries to government officers working standard hours in an office.

“*I am getting the same pay as any government officer in bank, taxation or teaching in school with same pay grade. What’s the point in working so hard, attending 24-h calls, working on all public holidays, disasters, pandemics when you can’t provide your family a better lifestyle or education than others?*”.(DH3)

The lack of differential reward, with surgery/anaesthesia consultants working day and night receiving the same wages as dermatologists working a few hours daily, engendered frustration and perceptions of injustice. Some suggested that if government could not change pay grades, at least inter-grade categories should be created, e.g., enabling anaesthesiologists to earn an amount per case alongside their regular salary. This would increase interest in public service and thus increase the workforce. Several suggested providing benefits instead of salary increases, as army officials received.

“*If not for all doctors, at least consultants should be offered housing, transport facilities, clubs and specific schools for children, or at least special quota seats for healthcare professional’s families along with their regular salary. This could overcome our workforce deficiency, as nobody wants to move out of their native country if they can get the best for their families here*”.(DH5)

All agreed that anaesthesiology required urgent attention to ensure appropriate incentives to attract sufficient workforce, but incentives should be dependent on qualifications (e.g., FCPS/MS should receive the highest incentives, MCPS midrange, DA lower, allied personal lowest). While the government started providing incentives to anaesthesiologists 20 years ago, amounts did not increase and became meaningless. Some suggested offering packages (e.g., including salary, facilities, housing, fuel) to anaesthesiologists according to their qualifications. Moreover, timely promotions and incentives could improve confidence and ultimately improve workforce quality and performance.

#### 3.2.5. Evening Privatization of Public Hospitals

Several participants suggested allowing private consultations in government hospitals during evening hours, as this could not only stabilise quality and price but improve equity among departments by giving equal chances and wages to qualified personnel. 

“*It was common practice in the past in the biggest tertiary care hospitals […]. Richest people would opt for state-of-the-art private wards in government hospitals as they were reliable*”.(TH3)

Participants agreed that private practice in government hospitals could reduce unqualified practitioners and improve wages for clinicians and allied staff by allowing patients to choose their anaesthesiologists, as private facilities normally did not have patients meet anaesthesiologists prior to surgeries.

“*Autonomous hospital bodies have the legislative authority to start private practice, but they do not want to take responsibility, as evening private practice needs cleanliness, up-to-date or at least decent waiting venues etc. Although 30% of income received from private patients goes to management for maintaining these things, they are not willing to burden themselves*”.(TH10)

Participants suggested that starting private practice in evenings where they served in the morning would not only promote quality but also improved relationships between doctors and their workplace, as they would want to make the facility welcoming and comfortable for themselves and their patients, which would also improve public trust.

### 3.3. Private Sector Health Facilities Obstacles

#### 3.3.1. Inadequate Salary and Facilities

Most participants suggested that private sector ‘monopoly’ (e.g., in which employers had sufficient market control to decide anaesthesiologists work and career conditions) was primarily responsible for driving qualified anaesthesiologists to emigrate from Pakistan and reported several ways the field had worsened. Most private hospitals listed a senior consultant-anaesthesiologist, while instead junior house officers, medical officers, or even operation theatre assistants provided anaesthesia due to insufficient quality-control or consequences.

“*Private sector is making a fool of the public by making modern buildings, interiors, reception private rooms etc., but as the public can’t enter operating rooms, the situation is grave there. Insufficient and out-dated monitoring and equipment, lack of proper sterilization, even drugs sometimes. So, complication rates are 60–70% more than in government hospitals…*”.(DH4)

Participants mentioned a few private healthcare companies that paid anaesthesiologists well but were controlled by a group of senior anaesthesiologists unwilling to allow anyone except their ‘favourites’ to join. Other anaesthesiologists had to choose from poor-quality hospitals and fixed remuneration.

“*In the private sector, not only are wages lower but also qualifications give you no edge. If they want to give an anaesthetist 5,000 for a case they would get one, whether it’s some OTA [Operation Theatre Assistant], HO [House Officer], MO [Medical Officer] or technician. They would give you no preference or better wage for your qualification, which not only reduces quality of anaesthesia but also reduces opportunities for skilled personnel in-country*”.(DH8)

Despite differences in age and experience, participants insisted if at least 20–25% of total operative charges were fixed for anaesthesiologists it would not only improve anaesthesiologist interest in private practice but also ensure qualified anaesthesiologists as facilities would lose the incentive to hire unqualified people if paying fixed rates. Several participants suggested that a membership organization working for anaesthesiologists must introduce a minimum wage, based on qualifications, and ensure none agreed to work for less than this set amount. One suggested an online portal, in which doctors enter each case they performed, to monitor participation of qualified personnel and mitigate misuse of credentials.

#### 3.3.2. Surgeon Dependency

Practicing consultants explained that anaesthesiologists were usually recruited through surgeons, especially in smaller facilities. As surgeons brought in cases, they were major contributors to private hospital wealth, so surgeons preferred the cheapest or most cooperative rather than most-qualified.

“*The worst thing in Pakistan’s private sector for anaesthesiologists is surgeons. You can practice only if you are connected to some surgeon who can call you for his surgery on his terms. If anaesthesiologists are directly collaborating with hospitals instead of surgeons, only then could checks-and-balances be kept*”.(QH4)

Most participants claimed that due to negligible anaesthesia mortality rates owing to improved equipment and drugs, surgeons assumed anaesthesia was just the injection and tried to save money by hiring less qualified anaesthesiologists. Surgeons assumed they could handle anaesthesia complications themselves as surgery was harder, though grave anaesthesia-related complications could happen in seconds. Several suggested that anaesthesiologists should be employed by every private hospital, not just reputable ones, instead of being on-call for surgeon-dependent cases. Thus, surgeons could not try to reduce costs by hiring less qualified anaesthesiologists. 

“*With my more than 19 years of experience in this field, I can assure that the only thing to overcome this issue is to make a rule that anaesthesiologists should meet with patients two days before surgery, for pre-operative assessment and rapport-building or patient get to choose an anaesthesiologist himself instead of the surgeon or hospital. This is the only way this malpractice could be reduced*”.(QH1)

#### 3.3.3. Lack of Out-of-Theatre Practice

Most participants claimed their non-operative practice was insufficiently supported by other specialties.

“*Chronic pain management is a definitive branch of anaesthesia, but it has no scope in private practice as fellow consultants of oncology or ortho would never refer them to any anaesthesiologist. They think they can handle everything by themselves*”.(DH6)

As anaesthesia is not curative, nobody comes to hospital looking for anaesthesia. Thus, anaesthesiologists had minimal patient interaction and needed other consultants to refer pain patients to them. However, financial interests prevented most colleagues from doing so. Most participants described this as a major disincentive to work in Pakistan, as they had little chance of lucrative private practice. All participants advocated legislation to define specialty roles and end ‘one-man shows’ in surgery.

#### 3.3.4. Lack of Surgery Categorisation

All participants insisted that hospitals should categorise surgeries to ensure enough qualified anaesthesiologists per surgery, particularly private hospitals that often only employed one anaesthesiologist rather than a team and no senior could be called for help. For example, several suggested that severest ‘Category A’ cases (i.e., cardiac, fire arm injury, road traffic accident, pulmonology, transplant) should only be handled by FCPS/MS anaesthesiologists, category B (i.e., laparotomy, open fractures) by MCPS/DA anaesthesiologists and category C (i.e., amputations, caesarean sections) by residents or medical officers.

“*Strict legislations are required for private sector as they take anaesthetists for granted. Not every anaesthetist is capable of handling any case or any kind of complication, so they must be called according to the type of surgery*”.(QH7)

## 4. Discussion

This study is one of the first to examine the perceived challenges anaesthesiologists experience in government and private hospitals in Punjab, which has reduced their trust in the health system and encouraged them to emigrate for better remuneration and healthier working conditions, creating a ‘brain drain’ that weakens service provision. Anaesthesiologist shortages are common in lower-income countries due to the similar emigration of qualified staff [25]. The limited anaesthesia workforce in Pakistan has been described as ‘a crisis’ [26] that must be addressed.

Physical security was a major concern among anaesthesiologists in public hospitals, aligning with findings in other countries [27,28,29]. For example, Indian health professionals experienced similarly frightening assaults by patients’ relatives in emergency departments [14,30,31]. This insecurity affected anaesthesiologists, other professionals, and patients as referrals due to fear of backlash could result in increased morbidity and mortality because of the time required in reaching distant alternative hospitals. A second major concern, in both public and private hospitals, was remuneration [32]. Javed et al. found 77% of anaesthesiologists in Pakistan’s public sector were dissatisfied with their income while 52% in the private sector were also dissatisfied [33]. Perceived disparities were also described in other LMICs as anaesthesiology depends on surgeons and is thus may be poorly remunerated [34]. Another study in India described multiple challenges, including dependency on surgeon selection in private practice [35]. A third issue, under which most complaints among anaesthesiologists in both public and private sectors could be summarised, was perceived unfairness/lack of opportunities compared with other clinical specialties. This is reflective of the international literatures on performance-based and education-based incentives [36], suggesting that retaining anaesthesiologists’ interest requires more than fixed or seniority-based salaries and benefits [37,38,39].

While anaesthesiologists suggested various ways to mitigate their concerns, including minimum salaries, incentives/benefits, greater influence within hospitals, not all suggestions may be equally feasible or even desirable. For example, privatising government hospitals in evenings has been shown to reduce public-sector efforts as providers shift patients to private-sector hours to be able to charge for their services [40].

Finally, our gendered findings were complex. Challenges reported among women anaesthesiologists, such as gendered insecurity/domestic responsibilities and insufficient support for childcare, have been reported by working women across sectors and countries [41]. In higher-income countries more women can mitigate gendered concerns with higher salaries and, thus, unlike many female anaesthesiologists in Punjab, they can still choose to work [25,26,27,42]. 

### 4.1. Implications for Policy and Practice

It appears likely that improved salaries or incentives will be needed to retain skilled anaesthesiologists in Punjab, particularly in government hospitals [36]. Salary legislation could help Pakistan ensure participation of qualified anaesthesiologists, with annual increases as described in previous research on anaesthesiologist incentivisation [39]. Lessons can be drawn from the literature on performance-based incentives, to ensure incentivisation is based on performance and qualifications rather than tied to seniority and professional connections [37,38,39]. However, any salary or incentive structure changes will need to be carefully considered to avoid creating perverse incentives and to ensure sustainability.

Greater private sector regulation is likely needed in the long term, given many complaints related to perceived lack of adequate care provision and quality in the private sector. Partial privatisation of government hospitals, e.g., in evenings, also requires careful assessment to avoid worsening access equity for patients. However, it could increase participation among qualified anaesthesiologists and help overcome private-sector monopolies, as patients reportedly have more trust in privatised government facilities than fully private hospitals [43]. While task-shifting to lower-skilled cadres for less complex cases could seem a possible solution, as described by Mavalankar et al. in other South Asian countries, such approaches are neither legal nor well regarded in Pakistan [4]. 

Women’s participation is increasing in every medical speciality in Pakistan, including anaesthesia, a trend that can be seen in many countries [40]. However, more efforts are needed at institutional or government levels, particularly to ensure sufficient security and trustworthy childcare, if women’s greater participation is to be sustainable.

Additional institutional and individual focus should be given to improving the surgeon-anaesthesiologist relationship during training, which would increase mutual respect for and understanding of each other’s domains and potentially improve monetary negotiations [44,45,46]. However, such shifting of professional norms requires that anaesthesiology be better recognised so anaesthesiologists can advocate for themselves, e.g., on the importance of preoperative assessments and conducting these at least 2–3 days before scheduled surgery to help develop rapport with patients as surgeons do. The National Society of Anaesthesiologists could work to ensure rights of anaesthesiologists in the private sector, minimum acceptable or percentage wages according to qualifications, and reporting of malpractice in private hospitals and push for more out-of-theatre exposure [47,48]. 

Globally, high-income countries must consider the damage international recruitment of anaesthesiologists from low-income countries can inflict on already-constrained health systems and either contributes substantively to their training costs or train enough qualified personnel in their own countries.

Further research could include comparative interviews with surgeons and non-consultant anaesthetists to gain a more comprehensive view of concerns raised by anaesthesiologists, implementing and assessing performance-based incentive interventions, and examination of anaesthesiologists’ experiences in other subnational regions.

### 4.2. Limitations

Several limitations should be considered. First, only anaesthesiologists’ perspectives were included, and other specialties or patient perspectives might have provided additional insights. Second, participants were all from Punjab province, which has relatively high socioeconomic, educational, and health provision indicators and anaesthesiologists’ experiences in other provinces may be worse. Third, SS conducted this as part of her PhD studies—with imposed time and funding constraints—and is relatively new to qualitative research, so some nuances may have been missed. However, she worked with experienced researchers to minimise this.

## 5. Conclusions

Fundamentally, anaesthesiology needs better recognition in Pakistan. Without this, any demands or suggestions for retention are unlikely to succeed. Perceived challenges and needs among anaesthesiologists must also be considered within this process. Public–private interlinkages and inequities likely require additional regulation, while surgeon dependency and underpayment should be reduced, e.g., by ensuring a fair wage or remuneration percentages by type of surgery. Anaesthesiologist-patient and anaesthesiologist-surgeon relationships must be improved, in both private and public hospitals, e.g., through public awareness and demand-generation. Performance-based incentives could be tested, along with institutional adaptations to address insecurity and gender inequities. Addressing such issues would increase the chances that anaesthesiologists could stay and have a fulfilling career in Pakistan.

## Figures and Tables

**Table 1 ijerph-19-13427-t001:** Participants’ characteristics.

ID	Job	Gender	Age (in years)	Experience (in years)
TH1	Anaesthesia consultant at teaching hospital	Male	42	10
TH2	Anaesthesia consultant at teaching hospital	Male	54	15
TH3	Anaesthesia consultant at teaching hospital	Male	37	7
TH4	Anaesthesia consultant at teaching hospital	Female	36	5
TH5	Anaesthesia consultant at teaching hospital	Male	59	28
TH6	Anaesthesia consultant at teaching hospital	Male	44	11
TH7	Anaesthesia consultant at teaching hospital	Female	35	6
TH8	Anaesthesia consultant at teaching hospital	Female	55	24
TH9	Anaesthesia consultant at teaching hospital	Male	48	18
TH10	Anaesthesia consultant at teaching hospital	Male	50	19
DH1	Anaesthesia consultant at DHQ hospital	Female	31	2
DH2	Anaesthesia consultant at DHQ hospital	Female	36	3
DH3	Anaesthesia consultant at DHQ hospital	Male	40	6
DH4	Anaesthesia consultant at DHQ hospital	Male	51	18
DH5	Anaesthesia consultant at DHQ hospital	Male	45	9
DH6	Anaesthesia consultant at DHQ hospital	Female	42	5
DH7	Anaesthesia consultant at DHQ hospital	Male	37	8
DH8	Anaesthesia consultant at DHQ hospital	Female	33	4
QH1	Anaesthesia consultant at THQ hospital	Male	52	19
QH2	Anaesthesia consultant at THQ hospital	Male	48	5
QH3	Anaesthesia consultant at THQ hospital	Male	34	1
QH4	Anaesthesia consultant at THQ hospital	Male	41	6
QH5	Anaesthesia consultant at THQ hospital	Male	37	5
QH6	Anaesthesia consultant at THQ hospital	Female	42	4
QH7	Anaesthesia consultant at THQ hospital	Male	32	1

## Data Availability

Due to the qualitative nature of this study, the transcripts cannot be published to allow for the anonymity of participants. However, data are available from the corresponding author upon reasonable request.

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
