# Peer review of "Professional Obstacles to Anaesthesiology Practice in Punjab, Pakistan: Qualitative Study of Consultant Anaesthesiologists’ Perspectives"

_ijerph, 2022, doi:10.3390/ijerph192013427_

Round 1
Reviewer 1 Report
See attached file

Author Response
Please find attached a detailed response.

Reviewer 2 Report
Dear editors/ authors,
Thanks for submitting this article.
This is the first exploration of this topic - anaesthetist views on barriers to working in Punjab Pakistan and this has important implications for workforce there.
As a small qualitative study, methods and analysis were appropriate. The sample was recruited via snowballing and appears to have reached saturation.
There were some interesting findings - in particular the disparity of income between surgeons and anaesthetists, and risk of assaults on anaesthetists, and female preponderence of anaesthetists. Discussion/ conclusions about addressing these issues was appropriate. Discussion re need for safe environment for anaesthetists, for example security, or female specific on - call rooms, also need for education of the public about risks of anaesthesia, was appropriate.
Discussion re need for time for consultation with patients prior to surgery was also appropriate.
Discussion re conducting private anaesthesia at night after doing public anaesthesia in the daytime, may not be a practical solution. However expanding the workforce for example using technicians for less complex cases may be possible pending funding.
It would have been interesting to know if the interviewees reported times when critical surgery did not proceed because no anaethetist was available.
Anaesthetists being appointed by surgeons is a common phenomenon, so the recommendation of working independently may not be practicable, - perhaps salaried anaesthetists who work with whichever surgeon is operating would be a solution. Alternatively, anaesthetics being able to charge the fees they wish to charge could be an alternative.
The authors have not include a sentence regarding developed nations recruiting anaesthetists for underserved nations and that training sufficient numbers of anaesthetists in developed and other nations is also critical.
Overall, the language was a little stilted and a further brief review for English language would be useful.
Author Response

(The authors gave the same response as above.)

Reviewer 3 Report
Dear authors,
I would like to congratulate all of you for your hard research work! Starting with the selected topic, it is one of great interest, because the retention of skilled medical personnel remains a major challenge for all developing countries around the world! The hard pandemic times have accentuated such vulnerability even more!
In general terms, please find mentioned below the most important aspects which draw my attention:
- The paper is very well structured and also very clearly presented, so it is very easy to follow the main idea of the manuscript! I really appreciate the special section concerning the policy implications of the study, because the research definitely addresses some important obstacles for anaesthesia specialists which must be resolved. So the therapeutically measures proposed have a high relevance for public health sector, in general and could inspire sound policies in this respect.
-The abstract is properly conceived, highlighting the most important aspects of the entire work!
-The introduction states very clearly the research objective, as well as the novelty of the analysis.
-The method is very well presented, consequently brings additional added value to the research, by providing to the audience all the necessary details in order to better understand how results were obtained.
- Considering the results and discussion section, I have really appreciated the fact that the authors have chosen (and they did it very well) to separate the obstacles from the public and private hospitals which anaesthesia workforce is confronted with. The argumentation line is excellent!
- Conclusions are clear and fully validate the research results!
Congratulations and many thanks for the opportunity to read such an interesting paper!
Author Response

(The authors gave the same response as above.)
